# Evaluation of the Effect of Gold Mining on the Water Quality in Monterrey, Bolívar (Colombia)

**Alison Martín [1,2], Juliana Arias [1], Jennifer López [1], Lorena Santos [1], Camilo Venegas [1]****, Marcela Duarte [3], Andrés Ortíz-Ardila [4], Nubia de Parra [3], Claudia Campos [1] and Crispín Celis Zambrano [2,*]**

[1]   Department of Microbiology, Laboratorio de Indicadores de Calidad de Agua y Lodos (LIAL), Pontificia Universidad Javeriana, Carrera 7 No. 43–82, Bogotá 110231, Colombia; alison.martin@javeriana.edu.co (A.M.); julichaque@gmail.com (J.A.); jetalomo@gmail.com (J.L.); lorena28sant@gmail.com (L.S.); c.venegas@javeriana.edu.co (C.V.); campos@javeriana.edu.co (C.C.)

[2]   Department of Chemistry, Pontificia Universidad Javeriana, Carrera 7 No. 43–82, Bogotá 110231, Colombia

[3]   Independent Investigator, Bogotá 110111, Colombia; duartemarce@yahoo.com (M.D.); nubiapdeparra@gmail.com (N.d.P.)

[4]   Hydraulic and Environmental Engineering Department, Pontificia Universidad Católica de Chile, Av. Vicuña Mackenna 4860, Macul, Santiago 7820436, Chile; adortiz@uc.cl

\*   Correspondence: crispin.celis@javeriana.edu.co

**Abstract:** Gold mining uses chemicals that are discharged into rivers without any control when there are no good mining practices, generating environmental and public health problems, especially for downstream inhabitants who use the water for consumption, as is the case in Monterrey township, where the Boque River water is consumed. In this study, we evaluate Boque River water quality analyzing some physicochemical parameters such as pH, heavy metals, Hg, and cyanide; bioassays (*Lactuca sativa, Hydra attenuata,* and *Daphnia magna*), mutagenicity (Ames test), and microbiological assays. The results show that some physicochemical parameters exceed permitted concentrations (Hg, Cd, and cyanide). *D. magna* showed sensitivity and *L. sativa* showed inhibition and excessive growth in the analyzed water. Mutagenic values were obtained for all of the sample stations. The presence of bacteria and somatic coliphages in the water show a health risk to inhabitants. In conclusion, the presence of Cd, Hg, and cyanide in the waters for domestic consumption was evidenced in concentrations that can affect the environment and the health of the Monterrey inhabitants. The mutagenic index indicates the possibility of mutations in the population that consumes this type of water. Bioassays stand out as an alert system when concentrations of chemical contaminants cannot be analytically detected.

**Keywords:** bioassays; gold mining; health risk; mercury; microbiological indicators; mutagenicity; toxicity

## 1. Introduction

Gold mining in developing countries is the main source of income for 30 million miners globally. About 12% of global gold production is through illegal mining that provides a significant economic benefit to miners but also proves hazardous/harmful for the environment by causing impacts such as water source sedimentation, land cover degradation, deforestation, soil degradation, and chemical contamination with mercury, cyanide, nitric acid, and zinc [1–3]. In Colombia, despite the various alternatives to avoid the use of Hg in gold extraction, the use of the elemental Hg–Au amalgamation method in small-scale artisanal mining areas is extensive [4,5].

Within the gold mining protocols, mercury and cyanide play an important role. These materials are easy to use, available at a low-cost, and easily accessible. However, there is little awareness among

the users or villagers about the use risk of cyanide and mercury in the gold extraction process [1,2]. This activity has led to serious pollution of terrestrial and aquatic ecosystems in emerging countries, impacting mining and fishing communities, and also these polluting elements can reach human beings [4–7].

According to the records of the Colombian Mining Association (ACM), gold production increased in 2020, going from 8.9 tons in 2019 to 9.5 tons in the first quarter of 2020, this represents a growth of 7% [8]. On the other hand, for gold extraction, 86% is considered illegal, taking place without a recognized mining title or without being registered. Medium-scale mining constitutes up to 26% and large-scale mining only takes up 2% of the total [9,10]. The population of the Bolívar department is 2,195,811 inhabitants, according to DANE's (National Statistics Administrative Department) projection for 2019 [11]. According to Carranza-Lopez et al. [4] the gold-mining districts (GMDs) at the department of Bolívar have extensive Hg contamination, and this situation requires special attention to reduce environmental and human health problems.

Municipalities of Montecristo, Santa Rosa del Sur, San Martín de Loba, Morales, San Pablo, Barranco de Loba, and Simití that are in Bolívar are where gold mining mainly takes place. Simití is known as the municipality that has the largest gold mining activity within the Bolívar department. It has an estimated population of 10,360 inhabitants in an area of 1345 km$^2$, the mining activity occurs in the Boque River, which flows in Simití. It starts on Serranía San Lucas, passes through Monterrey district, and flows into Magdalena River [12–14] (Figure 1).

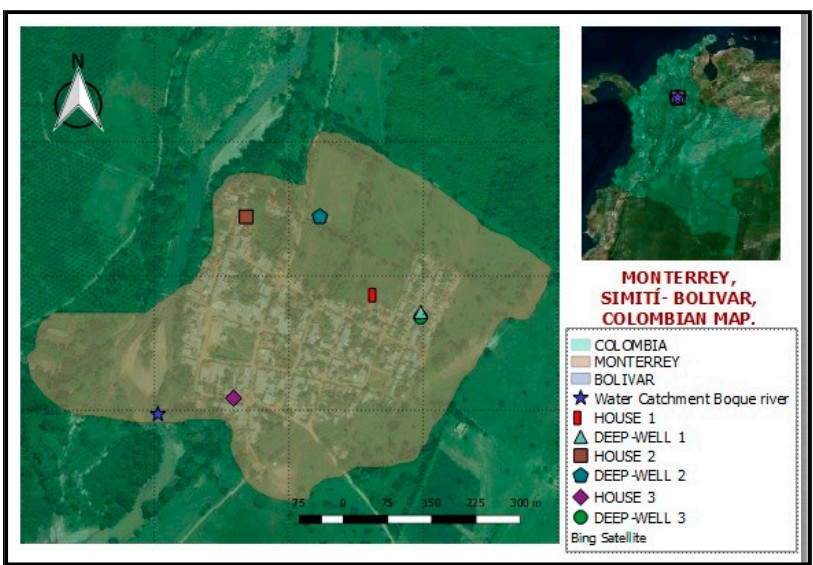

**Figure 1.** Location of sampling stations in Monterrey, Simití-Bolívar, Colombia. Source: authors.

Gold mining in Middle Magdalena has been carried out through artisanal practices, without considering the implications in the community and ecosystems due to the practice of non-regulated techniques affecting the environment, natural resources as well as health conditions and welfare of the population. Gold mining severely affects water resources, biodiversity, animals, flora, and fauna in its geographical area. In addition, the presence of certain types of mining settlements bring to pass certain types of domestic wastewater discharges without treatment to the Boque River, affecting the quality of the water and the inhabitants downstream [15–19]. The discharge of wastewater into a water body involves a large number and diversity of heavy toxic chemicals, many of which are unknown. These chemicals may react with each other, which can increase the toxicity level, which creates a negative impact on the structure and functioning of the natural ecosystem [4,20].

To determine the effect of gold extraction in the region, the evaluation of physicochemical parameters of the water is required. Nevertheless, the illegal settlements do not have sanitation

systems, so microbiological contamination becomes an additional problem. However, even if in some of the above-mentioned situations the parameters could be between the legal requirements, it should be considered that trace heavy elements might have an impact on the population and the ecosystem after long periods of exposure. Thus, it is necessary to test different representatives of the trophic chain to identify the impact of the pollutants through bioassays tests [20–23].

Bioassays are described as alert mechanisms for long-term periods of exposure to chemical pollutants. These are used as indicators of substances that are harmful to living cells and tissues, useful even in the cases where physicochemical parameters fulfill the requirements of water quality [24]. Likewise, this possible bioaccumulation of chemical elements in the trophic chains can generate mutagenicity or toxicity, which is why it is important to be able to establish whether a complex system such as a water sample from a mining region has these undesirable characteristics, which can be detected by the Ames test or bioassays [25–27]. Some Latin American countries have made progress in the application of toxicity tests, while for Colombia, toxicity tests in natural environments are scarce compared to the evaluation of hazardous waste and industrial dumping [28,29]. On the other hand, the Ames test has proven to be effective for the identification of potentially carcinogenic or mutagenic chemicals, achieving its immediate adoption and its requirement by regulatory authorities around the world [30]. In the Ames test, *Salmonella typhimurium* (*S. typhimurium*) is used as an indicator of bacterial mutagenesis as a consequence of exposure to chemical contaminants [25,26].

Taking into consideration that the water of the Boque River is used in human consumption without treatment and it collects chemical pollutants from the mining activity, such as mercury and cyanide, the use of the Ames test in the evaluation of this water will permit the evaluation of its possible carcinogenic or mutagenic effect, making it a relevant issue for the inhabitants of Simití. For this reason, it is necessary to have data on bioassays and Ames test indicators in environmental samples, especially in mining, which has become one of the most important fonts of economic resources in Colombia and at the same time of damage not sufficiently evaluated to date. In order to have a complete evaluation of the water quality in relation to the possible presence of bacteria, viruses, and parasites and the risk to the inhabitants, it is necessary to use indicators of fecal contamination, with the most used indicators being total coliforms and *Escherichia coli* as bacterial indicators and somatic phages as viral indicators, which allow indication of the presence of pathogenic microorganisms in the water.

The aim of this research is to evaluate the impact generated by the exploitation of bad mining practices such as the use of dangerous chemical compounds in gold mining, which are drained into surface waters such as the Boque River in the South of Bolívar, Colombia, as well as the waste generated in the mining settlements. The assessment of the impact on the environment, living organisms, and human health will be done through the detection of heavy metals, microbiological indicators, and bioassays, which through a joint assessment will provide important aspects to protect the health of the inhabitants of Monterrey.

## 2. Materials and Methods

### 2.1. Study Area

The Boque River has an approximate area of 876 km$^2$ and merges into the Magdalena River. The Monterrey district belongs to the Municipality of Simití, Department of Bolívar (Colombia). The inhabitants who live within the Monterey township collect and use the water from the Boque River for different activities, this being the main source of water supply (Figure 1) [27].

### 2.2. Water Physicochemical Analysis, Heavy Metals, and Cyanide Detection

Some physicochemical parameters were analyzed such as pH (pH/T tester pHep®4, Hanna Instruments, RI, USA) [31], chemical oxygen demand (COD photometer Hanna Instruments, New England, RI, USA) [32], and total solids by the gravimetric method [33]. The detection of heavy metals was performed using a Varian SpectrAA 220 G Atomic Absorption Spectrometer(Varian-Agilent

Inst., Palo Alto, San Francisco, CA, USA), following previous publications: cadmium, chromium, zinc, and nickel [34]; mercury in a direct mercury analyzer (DMA-80, Milestone Inc., Sorisole, Italy) [35]; and cyanide in a portable photometer (Hanna Instruments, RI, USA) [36]. All reactive, analytical standards and reference materials were purchased from Merck (Merck KGaA, Darmstadt, Germany). The results obtained in the samples from Village Gato, Village Tigui, and the water catchment of the Boque River were compared with normative 0631/2015 [37], which establishes the parameters to be monitored and maximum permissible limits in the specific discharges of non-domestic wastewater (precious minerals and gold). While for houses and deep-well underground sites, they were evaluated in compliance with normative 2115/2007 [38], which regulates water for human consumption.

### 2.3. Bioassays

In the bioassays, *Lactuca sativa (L. sativa)* [39] and *Hydra attenuata (H. attenuata)* [40] were used as a biological indicators of water quality. After the follow-up of the results of the two first collections of water samples, a modification of the protocol was performed replacing *Daphnia magna (D. magna)* [41] instead of *H. attenuata* due to no evidence of sensitivity against the possible harmful substances that might be present in the water samples by *H. attenuata*. The effects on organisms can be inhibition, sublethality, and lethality volume/volume (v/v). The water samples were diluted in four different concentrations 25%, 50%, 75%, and 100% (v/v); reconstituted hard water (160–180 mg/L $CaCO_3$) was used as diluent for the *D. magna* and *H. attenuata*; while for *L. sativa*, distilled water was used. The response of *H. attenuata* was read using a binocular stereoscope (Leica). Before taking the readings, the containers were shaken in a circular way to reactivate the movement of *D. magna* and confirm their state. In the case of *L. sativa*, graph paper was used to measure the length of the radicle.

#### 2.3.1. Endpoint and Toxic Response Model

*L. sativa*, half-maximal inhibitory concentration ($IC_{50}$): root growth reduction or inhibitory effects on lettuce seed germination and root growth after 5 days. *D. magna*, lethal concentration ($LC_{50}$): number dead/total number or lethal effects of water were observed after 48 h of exposure, and *H. attenuata*, median effective concentration ($EC_{50}$): density reduction or lethality test, produced by irreversible morphological changes after 96 h of exposure.

#### 2.3.2. Toxicity

To calculate the lethal concentration ($LC_{50}$), the half-maximal inhibitory concentration ($IC_{50}$), and median effective concentration ($EC_{50}$), the method used was Environmental Protection Agency (EPA) Probit analysis model [42–44]. When results in $EC_{50}/LC_{50}/IC_{50}$ cannot be reported by the statistical program, they are reported as the percentage of effect (%) in the lowest concentration at which the event is still present on the evaluated population [20].

### 2.4. Ames Test

The method was applied according to Ames [45] using *Salmonella typhimurium (S. typhimurium)* TA98 and TA100 strains to evaluate possible mutagenicity. The cultures were grown in Oxoid nutrient broth No. 2. The samples of water were diluted in four different concentrations 25%, 50%, 75%, and 100% (*v/v*). The mutagenic effect was evaluated from the number of revertant colonies per plate. The plates were prepared in triplicate for every test sample, and the result presented was the mean of triplicate observation. The mutagenic activity was detected after 120 h of exposure at 37 (±2) °C. The revertant colonies' readings were counted using an automatic colony counter (Industrial Scientific). For accuracy of the results, mutagenic index (MI) values greater than or equal to two (≥2) were considered mutagenic [46].

*2.5. Statistic Analysis*

To establish whether there is a relationship between evaluated physicochemical parameters and heavy metals with results of bioassays, a one-way analysis of variance (ANOVA) was performed with a significance level of $p < 0.05$.

*2.6. Microbiological Analysis*

The determination of total coliforms and *E. coli* as indicators of bacterial fecal contamination was performed according to the ISO 9308-1 standard method [47]. Cellulose acetate membranes of 0.45 μm × 47 mm (Sartorius) were used for the filtration. Dark blue/purple colonies on Chromocult agar (Merck) were presumed to be *E. coli*. The detection and quantification of somatic coliphages as indicators of viral fecal contamination was performed according to the ISO 10705-2 standard procedure and modified Scholten's agar (MSA) (OXOID) was used for the detection of coliphages [48].

The results obtained in the samples from Village Gato, Village Tigui, and the water catchment of the Boque River were compared with normative 1594/1984 [49], which regulates waters that can be treated by conventional systems for human consumption. While for houses and deep-well underground sites, they were evaluated in compliance with normative 2115/2007 [38], which regulates water for human consumption. Normative 2115/2007 only beholds the microbiological quality concerning total coliforms and *E. coli.* Although coliphage concentrations are not regulated within Colombian normatives, their detection is relevant since they confirm contamination of fecal origin and the possible presence of pathogenic viruses, both in drinking water and in water for human consumption.

## 3. Results

*3.1. Physicochemical Parameters*

Parameters such as COD, total solids, and pH did not exceed the limits of Colombian normative 0631/2015 [37], in the first three sample stations. However, the level of cadmium was excessive in the Village Gato station in the first sampling with a concentration of 0.05 mg/L. Chromium did surpass the limit in the first sampling in Village Tigui. Likewise, mercury in Village Gato exceeded the limit in the third sampling (0.0029 mg/L). Moreover, in the Village Tigui station, the measurements in the second sampling exceeded 0.0025 mg/L and those in the water catchment of the Boque River in the first sampling were also excessive with 0.0022 mg/L. Furthermore, the permitted concentration of cyanides in the first and second sampling with a concentration of 1.02 and 1.32 mg/L, respectively, was excessive at the Village Tigui sampling station. While in the water catchment of the Boque River station, the prescribed cyanide level was exceeded in the second sampling with a concentration of 1.57 mg/L (Appendix A—Table A1).

The values of heavy metals analyzed in the last two sampling stations were compared with normative 2115/2007 [38] based on waters for human consumption. On one hand, the level of cadmium exceeded the limits established by the regulations for the house station in the second and third sampling with concentrations of 0.03 and 0.01 mg/L, respectively. While the cyanide concentration was exceeded only in the second sampling (1.11 mg/L). On the other hand, the established values of cadmium exceeded only in the second sampling in deep-well underground, presenting a concentration of 0.01 mg/L. Mercury was detected in each of the samples for both the house and deep-well underground stations, but the concentrations did not exceed the limits established in normative 2115 of 2007 [38]. The other metals evaluated (Zn, Ni, and Cr) were not detected in any of the samples analyzed (Appendix A—Table A2).

*3.2. Bioassays*

Table 1 shows the different percentages of growth inhibition of *L. sativa* in three samples for five evaluated stations. As the results show, there was root inhibition in some sampling stations and overgrowth in others. In Village Gato, in the third sampling, there was excess growth at a concentration

of 25%. Likewise, in Village Tigui, the greatest inhibitions registered were observed at a concentration of 25%. In general, among all the sampled stations where the greatest inhibition was observed the highest was in Village Tigui, followed by house, deep-well underground, Village Gato, and the smallest recorded in the water catchment of the Boque River.

**Table 1.** Bioassays results with *Lactuca sativa*.

| Sampling Station (n = 15) | 1S % (*v/v*) Effect | 2S % (*v/v*) Effect | 3S % (*v/v*) Effect |
| --- | --- | --- | --- |
| Village Gato | 33% Inhibition to 75% | 27% Inhibition to 100% | 133% Growth to 25% |
| Village Tigui | 44% Inhibition to 25% | 24% Inhibition to 25% | 4% Inhibition to 50% |
| Water catchment of the Boque River | 0% Inhibition to 100% | 27% Inhibition to 100% | 110% Growth to 100% |
| House | 34% Inhibition to 25% | 19% Inhibition to 75% | 6% Inhibition to 50% |
| Deep-well underground | 35% Inhibition to 50% | 36% Inhibition to 75% | 6% Inhibition to 75% |

S: sampling, the numbers 1S, 2S, and 3S correspond to the months of July, September, and December in which the sample was taken; n: is the number of samples.

In Table 2, it is observed that the *D. magna* indicator has different mortality percentages since the same four concentrations of the sample are evaluated as in the *L. sativa* bioassay. The highest was 23% mortality at a concentration of 50% at the water catchment of the Boque River station and 17% mortality at 75% at the Village Tigui sampling station, followed by the house and deep-well underground stations. Finally, the lowest concentration of mortality was obtained in the Village Gato station with 33% mortality at 100%. For *H. attenuata*, it was not possible to determine the $EC_{50}$ and $LC_{50}$ values because there were no morphological changes, indicating lethality or sublethality, in the three samplings carried out, reporting 0% sublethality at 100% (*v/v*) and 0% lethality at 100% (*v/v*).

**Table 2.** Results of bioassay with *Daphnia magna*.

| Sampling Station (n = 15) | 3S % (*v/v*) Effect |
| --- | --- |
| Village Tigui | 17% mortality to 75% |
| Village Gato | 33% mortality to 100% |
| Water catchment of the Boque River | 23% mortality to 50% |
| House | 47% mortality to 100% |
| Deep-well underground | 43% mortality to 100% |

S: sampling, the number 3S corresponds to the month of December in which the sample was taken; n: is the number of samples.

### 3.3. Ames Test

The results obtained with the Ames test using *S. typhimurium* TA98 and TA100 strains are presented in Table 3, where the mutagenic index (MI) is shown for each condition used in the assay. According to Table 3, for the second sampling of the house station, mutagenic values were observed for both strain TA98 and TA100 in each of the concentrations evaluated. While for Village Gato, with strain TA100, a value of 2.4 was observed in the 100% concentration (Table 3). For the other sampling sites, there was no mutagenic index.

**Table 3.** Mutagenic index, for each concentration analyzed in the five sampling stations with strain *Salmonella typhimurium* TA98 and *S. typhimurium* TA100.

| Sampling Stations (n = 15) | Concentration v/v (%) | TA98 (MI) | | | TA100 (MI) | | |
|---|---|---|---|---|---|---|---|
| | | S1 | S2 | S3 | S1 | S2 | S3 |
| Village Gato | 25 | 0.77 | 1.07 | 0.00 | 0.97 | 0.46 | 1.00 |
| | 50 | 0.86 | 0.87 | 0.20 | 1.09 | 0.68 | 1.20 |
| | 75 | 1.06 | 1.49 | 0.60 | 1.27 | 0.77 | 1.70 |
| | 100 | 1.12 | 1.93 | 1.00 | 1.51 | 1.14 | 2.40 |
| Village Tigui | 25 | 0.97 | 0.70 | 0.00 | 1.37 | 0.20 | 0.60 |
| | 50 | 0.95 | 0.74 | 0.00 | 1.39 | 0.34 | 0.90 |
| | 75 | 1.12 | 1.14 | 0.10 | 1.48 | 0.33 | 1.00 |
| | 100 | 1.21 | 1.82 | 0.40 | 1.81 | 0.56 | 1.20 |
| Water catchment of the Boque River | 25 | 1.17 | 0.39 | 0.30 | 0.94 | 0.27 | 0.60 |
| | 50 | 1.39 | 0.63 | 0.50 | 1.00 | 0.24 | 0.70 |
| | 75 | 1.41 | 0.47 | 0.60 | 1.21 | 0.35 | 1.10 |
| | 100 | 1.50 | 0.59 | 0.90 | 1.71 | 0.44 | 1.30 |
| House | 25 | 0.68 | 41.78 | 0.40 | 1.01 | 11.05 | 0.60 |
| | 50 | 1.00 | 48.49 | 0.60 | 1.17 | 12.39 | 1.00 |
| | 75 | 1.06 | 56.62 | 0.80 | 1.36 | 13.75 | 2.00 |
| | 100 | 1.21 | 58.31 | 1.10 | 1.70 | 15.09 | 2.50 |
| Deep-well underground | 25 | 88.0 | 0.39 | 0.20 | 1.02 | 0.37 | 0.80 |
| | 50 | 0.88 | 0.50 | 0.50 | 1.16 | 1.21 | 1.20 |
| | 75 | 1.00 | 0.50 | 0.80 | 1.23 | 0.42 | 1.70 |
| | 100 | 0.55 | 0.64 | 0.90 | 1.28 | 0.64 | 2.00 |

S: sampling, the numbers 1S, 2S, and 3S correspond to the month of July, September, and December in which the sample was taken; n: is the number of samples; MI: mutagenic index.

### 3.4. Statistical Analysis

The statistical analysis performed to determine a possible relationship between the results of the physicochemical parameters against the toxicity indicators showed that there is a relationship between the inhibition of *L. sativa* concerning to mercury with a significance of $p < 0.05$.

However, due to the low number of samples analyzed for *H. attenuata* (three samples) versus the number of samples for *D. magna* (15 samples), it was not possible to establish whether there was a correlation with the concentration of metals or with the results of *L. sativa*.

### 3.5. Microbiological Analysis

Table 4 shows the results of the concentration of fecal contamination indicators (total coliforms, *E. coli*, and somatic coliphages) for the different types of water of the five sampling stations. Table 4 shows that concentrations between $10^3$ and $10^5$ colony forming unit (CFU)/100 mL for total coliforms were obtained at the different stations. While in the case of *E. coli*, concentrations between $10^3$ and $10^4$ CFU/100 mL were obtained. Somatic coliphages were detected in samples taken at Village Gato and house stations. Colombia does not have regulations for the presence of this indicator, although this is necessary since the presence of somatic coliphages represents a risk to the health of the community.

While, for the water catchment of the Boque River, Village Tigui, and deep-well underground stations, the presence of phages in some samples was not detected ($<1.0 \times 10^3$). The results were compared with decree 1594/1984 [49], which establishes the concentration of total coliforms ($2.0 \times 10^4$/100 mL) allowed in waters that will be treated by conventional systems: while normative 2115/2007 [38], establishes that the concentrations for total coliforms and *E. coli* for drinking water is 0 CFU/100 mL.

**Table 4.** Results of total coliforms, *Escherichia coli*, and somatic coliphages in waters of the Boque River and drinking waters.

| Sampling Stations (n = 15) | Microbiological Indicators | | | | | | | | |
|---|---|---|---|---|---|---|---|---|---|
| | Total Coliforms CFU/100 mL | | | E. coli CFU/100 mL | | | Somatic Coliphages PFU/100 mL | | |
| | 1S | 2S | 3S | 1S | 2S | 3S | 1S | 2S | 3S |
| Village Gato | $7.0 \times 10^3$ | $1.1 \times 10^5$ | $4.1 \times 10^5$ | $1.0 \times 10^3$ | $4.0 \times 10^3$ | $8.0 \times 10^4$ | $1.4 \times 10^3$ | $1.0 \times 10^2$ | 4.5 |
| Village Tigui | $1.3 \times 10^5$ | $3.2 \times 10^4$ | $2.2 \times 10^5$ | $2.0 \times 10^3$ | $2.0 \times 10^3$ | $3.0 \times 10^4$ | $4.9 \times 10^3$ | $3.0 \times 10^2$ | $<1.0 \times 10^3$ |
| Water catchment of the Boque River | $2.4 \times 10^4$ | $1.7 \times 10^4$ | $4.0 \times 10^5$ | $1.0 \times 10^3$ | $1.0 \times 10^3$ | $1.0 \times 10^4$ | $<1.0 \times 10^2$ | $<1.0 \times 10^3$ | 1.9 |
| House | $4.0 \times 10^4$ | $1.4 \times 10^5$ | $2.8 \times 10^5$ | $1.0 \times 10^3$ | $3.0 \times 10^4$ | $3.0 \times 10^4$ | $1.0 \times 10^2$ | $2.0 \times 10^4$ | $1.0 \times 10^2$ |
| Deep-well underground | $3.2 \times 10^4$ | $6.0 \times 10^4$ | $3.8 \times 10^5$ | $1.0 \times 10^3$ | $4.0 \times 10^4$ | $4.0 \times 10^4$ | $2.0 \times 10^2$ | $<1.0 \times 10^3$ | 1.0 |

CFU/100 mL: colony forming units in 100 mL of analyzed water; PFU/100 mL: plaque forming units in 100 mL of analyzed water; n: number of samples analyzed. S: sampling, the numbers 1S, 2S, and 3S correspond to the month of July, September, and December in which the sample was taken; <: less than the limit of quantification; n: is the number of samples.

## 4. Discussion

### 4.1. Bioassays

#### 4.1.1. *Hydra attenuata* and *Daphnia magna*

By applying the *H. attenuata* toxicity test, it was not possible to determine lethality or sub-lethality since there were no morphological changes in the three samples taken. An important factor that could influence why *H. attenuata* was not sensitive to contaminants present in this water, is that the toxicity of metals is modified by abiotic factors such as hardness, pH, and water temperature [50]. For example, if water hardness is high, the formation of metal complexes tends to increase, which in turn lowers the effect of toxic divalent metals [50,51].

*H. attenuata* has a higher sensitivity to toxic substances at acidic pH, compared to that at alkaline or neutral pH [52]. The pH value of the water sample from the Boque River is about 7 (Appendix A—Tables A1 and A2), which could influence *H. attenuata* not presenting sensitivity when heavy metals, cyanides, or other toxic substances are in the water.

Due to the results obtained with *H. attenuata* in the first two samples, *D. magna* was used in the last sampling, to find an animal indicator that presented a greater sensitivity to these types of contaminants. Table 2 shows different mortality percentages that were found, demonstrating the sensitivity of this organism to the contaminants present in the water of the Boque River that is consumed by the Monterrey population. Studies conducted by Forget et al. [53] with *D. magna*, show percentages of toxicity up to 70% against heavy metals. Castro-Català et al. [54] evaluated the toxicity of sediments and water in rivers with the presence of pesticides and heavy metals using *D. magna* as an animal indicator, showing that it can be sensitive to these types of samples, due to its high metabolic rate [55]. Likewise, Lattuada et al. [56], in southern Brazil, used *D. magna* as an indicator of toxicity in waters affected by coal mining, in which heavy metals such as Fe, Mn, Zn, Ni, Cd, and Pb were found. The results showed sensitivity by this indicator in this type of water and suggest the evaluation of toxicity in waters from gold mining. Moreover, studies conducted in China by Wu et al. [57] demonstrated that the most frequently encountered heavy metals in a region affected by gold mining were mercury and cadmium, as observed in the results found in the drinking water of the population of Monterrey (Appendix A—Tables A1 and A2).

#### 4.1.2. *Lactuca Sativa*

The differences observed between chemical parameters and toxicity may be related to the fact that the samples were not collected simultaneously and that it is not the same water because along the river route and on the different sampling days, diverse factors can alter its quality. Likewise, dilution effects due to rain, sedimentation, the introduction of new pollutants, among others, can have an influence.

Additionally, the water entering the treatment plant can be more contaminated, taking into account that it travels through tanks that are not in operation or comes into contact with sludge that might have a higher concentration of contaminants, which may return to the column of water.

The increase in germination, compared to the positive control (overgrowth), is related to the presence of organic matter because they are essential nutrients for *L. sativa* seed germination, and if they are available in high concentrations, they will stimulate growth. On the other hand, mercury and cyanide at the Village Tigui station (Appendix A—Tables A1 and A2) had higher concentrations. The inhibition rates of 24% and 44% for samples 1 and 2 at the concentration of 25%, affect the growth of the seed as the higher concentration of pollutants results in greater inhibition. Castillo et al. [24] found inhibition in the germination of seeds in waters contaminated with mercury and argue that it can occur due to the harmful effects caused by mercury at the cellular level in the seed. These results coincide with other studies where *L. sativa* has been proposed as a useful tool to evaluate and compare the toxicity of industrial effluents that present heavy metal contamination [58].

Likewise, the level of cadmium (Appendix A—Tables A1 and A2), also exceeded the minimum values established by the regulations for water for human consumption; studies have been reported where the exposure of *L. sativa* to this metal causes toxic and harmful effects that decrease its growth as the concentration of Cd, and thereby its adsorption, increases [59–61]. Just as the presence of heavy metals and cyanide has toxic implications in the plant and animal model, in the same way, it will affect the health of the human being [62,63].

Cd is one of the most toxic elements to which man is exposed since the accumulation of this metal in the body is gradual and increases with age due to its long half-life, greater than 20 years [64]. This is why eating food or drinking water with very high levels of cadmium causes severe stomach irritation, which causes vomiting, diarrhea, and sometimes death [64]. Moreover, cyanide exceeded the allowed limits (0.05 mg/L) in one of the samples analyzed in one of the houses (1.11 mg/L). The guide values of the World Health Organization [65] establish that the concentration of cyanide toxic to humans is 0.07 mg/L. Exposure to this concentration or higher may cause inhibition of cell growth, thereby affecting the breathing process and the metabolism of nitrogen and phosphate. It also inhibits the activity of some metalloproteins, joining cofactors such as the heme group of hemoglobin [66].

Finally, cyanide has acute effects on human health such as irritation of the eyes, nose, and throat. High exposure causes intoxication with headache, weakness, nausea, strong heartbeat, coma, and even death. As for chronic effects, it causes nosebleeds and nose lesions and can cause enlargement of the thyroid gland, which can interfere with its regular function [67].

### 4.2. Ames Test

In some cases, there was a decrease in reverts with increasing doses, which may be due to the presence of toxic substances that prevent the growth of bacteria [68]. However, in most of the sampling stations, a direct relationship was observed between the number of revertants and the increase in the concentration of heavy metals. This demonstrates the high probability of the presence of substances such as heavy metals and organic compounds in the Boque River that cause base-pair mutations and changes in the DNA reading frame of bacteria.

When observing the reversion of the strains, it was evidenced that they exceed 2–40 times the value of the negative control for the TA98 strain in the second sample in one of the houses, and from 2.0 to 2.5 for the TA100 strain in the third sample for the house and underground well. According to Orozco and Zuleta [69], some samples can exceed 100 times the negative control and these results are related to the quality of the water.

Likewise, Meléndez et al. [70] investigated the mutagenic activity of drinking water before and after chlorination at the Villa Hermosa plant, Medellín, Colombia, finding that contamination and chlorination influence mutagenicity. They used the Ames test with strains TA100 and TA98. Sierra et al. [71] evaluated the mutagenic activity of the Cauca River water with the same strains with and without the enzyme activator S9, finding that the highest rate of mutagenicity was observed with

strain TA98 without enzyme activator. However, the TA100 strain is characterized by presenting the *hisG46* mutation and has specific markers that give it greater sensitivity to the test; within these are the *uvr* mutation, the *uvrB* mutation, and the plasmid pKM101 [26,68].

Mesquidaz et al. [72] reported alarming figures in the mercury concentrations used in the gold extraction process in a mine in northern Colombia, which ranges from 50 to 100 tons in 2007. Furthermore, it is reported that, thanks to this pollutant present in water, the health of the population has been affected, since Hg was found in human hair at a concentration of 12.8 μg/g, a figure that is well above international standards. It has been shown that inhabitants of different municipalities in southern Bolívar where gold mining takes place have high levels of Hg contamination, and this situation requires special attention to reduce environmental and human health problems [4]. Mercury contamination has been linked to health problems, as direct absorption of mercury vapor released by incinerators in gold mining, or ingestion of mercury-containing wastes, causes hydrargyrism and poisoning. Mercury (Hg) is one of the heavy metals of greatest concern to populations that consume fish. This pollutant can be released from many sources and has various toxic effects in humans [73].

Some of the health problems caused are excessive salivation, shortness of breath and fatigue, bronchitis, tremors and irritability, personality changes (due to brain damage), a sensation of floating teeth and pain in them, kidney and respiratory damage that can lead to death from problems in the lungs and other organs of the body [74–76]. While breathing polluted air, elemental mercury can reach the brain, affecting nerve cells and the olfactory system. The main organs in which mercury accumulates are the brain and the kidney [77,78].

### 4.3. Statistical Analysis

The statistical analysis showed a relationship between the inhibition of *L. sativa* concerning mercury with a significance of $p < 0.05$; this inhibition in the germination of *L. sativa* with this metal was also reported in Chile, where the exposure of the seeds to Hg inhibited their growth [19]. The toxicity caused multiple harmful effects in the seed at the cellular level such as a change in permeability in the cell membrane and the affinity to react with phosphate groups and the sulfhydryl group (SH). When mercury interacts with the SH groups to form the S–Hg–S bonds, it disrupts the stability of the group can affect seed germination and seedling growth whose tissues are rich in SH groups [79].

On the other hand, it was not possible to establish a correlation between the vegetal and the animal model due to the number of samples collected. When comparing the results of the bioassays associating for *D. magna* and *L. sativa*, it was observed that the variability due to the sampling was not simultaneous for every sample and it could be possibly affected by a new spill in the river. Additionally, these are different organisms with different sensitivity to the contaminants present in the Boque River water and there exist other factors that can influence this response. For example, some bacteria can naturally modify mercury ($Hg^{2+}$) by ion methylation forming $CH_3–Hg^+$, which is more toxic and is incorporated into trophic chains, affecting the animal model more than the vegetable model [80].

### 4.4. Total Coliforms, Escherichia Coli, and Somatic Coliphages

The microbiological results confirm the high fecal contamination in all the sampling stations (Table 4). Total coliform concentrations exceeded the limits for Colombian regulations [49]. In the case of drinking water for human consumption (deep-well underground and house), total coliforms and *E. coli* were well above levels required by regulation for drinking water [38]. Likewise, in the case of drinking water for human consumption by the treatment system (Village Gato, Tigui, and water catchment of the Boque River), the concentrations allowed for total coliforms were exceeded [49].

Campos-Pinilla et al. [81] and Sánchez-Alfonso et al. [82] in studies carried out in the Bogotá River found a total coliform concentration between $10^3$ and $10^6$ CFU/100 mL and for *E. coli* between $10^3$ and $10^5$/100 mL. This coincides with the values found in this study, which range between $10^3$ and $10^5$ CFU/100 mL of total coliforms and for *E. coli* between $10^3$ and $10^4$ CFU/100 mL (Table 4). Likewise, studies conducted by Lucena et al. [83] and Sánchez-Alfonso et al. [82] in rivers show average

concentrations of somatic coliphages between $10^2$ and $10^4$ plaque forming unit (PFU)/100 mL, similar to those found in this study with ranges from 1 and $10^3$ PFU/100 mL. The concentration of microorganisms in river water varies depending on climatic factors, geographical area, and the amount of organic matter present in water bodies [84,85]. The mine exploitation site is a settlement space for the population that works in this activity legally or illegally, which generates a high level of household waste in the river causing contamination by the discharge of fecal matter and organic matter, which explains the concentration of indicators of fecal contamination. It is related to the absence of treatment systems and improper installation of septic tanks.

The detected concentrations of total coliforms and *E. coli* in all the drinking water samples and the detection of somatic coliphages in some samples of water used for human consumption confirm the fecal contamination and the possible presence of pathogenic viruses in the drinking water (Table 4). These concentrations of indicators are similar to those detected in river samples as reported by Lucena et al. [83], Campos-Pinilla et al. [81], and Sánchez-Alfonso et al. [82], which could increase the risk for residents.

## 5. Conclusions

The results obtained with the three toxicity indicators reveal that *H. attenuata* does not present sensitivity to toxic substances present in this type of water, so its use for this purpose is not recommended. On the other hand, *D. magna* showed sensitivity even in diluted samples as well as *L. sativa*, which showed growth inhibition and excessive growth in different concentrations of the analyzed water, inclusive of waters with pollutant concentrations below the detection level. The Ames test shows an increase in the revertants indicating the possibility of mutations in the population that consumes this type of water, which is correlated with the results of the mutagenicity test that showed a mutagenic effect in the five stations evaluated with both strains used in the study. The highest mutagenic index was found in the water sample taken from the house sampling station. The concentration of bacteria in the water exceeded the limits allowed by Colombian regulations, creating a health risk, also with an alert call to the presence of possible pathogenic viruses, and the risk that they imply for the inhabitants of Monterrey due to somatic coliphage levels determined.

This research recognizes the potential use of bioassays to evaluate the toxic effects generated by chemical wastes produced by gold mining and discharged into surface waters. The use of animal and plant models is recommended to evaluate said effects on the environment and public health and infer the damages that until now have not been sufficiently evaluated having as correlation factors physicochemical and microbiological parameters.

Finally, this research generated data that contribute to the knowledge of the effects caused to the environmental and public health by illegal and legal mining carried out with bad practices in emerging countries with inefficient controls of this type of activity. These assays used can help sanitation organizations in different countries to take preventive actions on this issue

**Author Contributions:** All authors contributed to all features of the paper. A.M., J.A., J.L., L.S., C.V., A.O.-A., and M.D. were involved in the sampling and analysis of bacteria, coliphages, bioassays, and physicochemical determinations. C.C. and C.C.Z. conceived the idea for the research and contributed to the development of the project by obtaining economic resources. C.C., N.d.P., and C.C.Z. provided consultation and interpretation of the results and contributed to writing the manuscript. C.V., A.M., and J.L. edited the manuscript. All authors have read and agreed to the published version of the manuscript.

**Funding:** This research and article publication was funded by Pontificia Universidad Javeriana, Bogotá, Colombia. Grant number 0005746 and 0008469.

**Acknowledgments:** The authors would like to acknowledgment the residents of Monterrey Sur de Bolívar, Colombia, and Programa de Desarrollo y Paz del Magdalena Medio by the logistical support in sampling.

**Conflicts of Interest:** The authors declare no conflict of interest.

## Appendix A

**Table A1.** Results of physicochemical parameter analysis compared with the normative 0631/2015.

| Physicochemical Parameters (n = 9) | Village Gato | | | Village Tigui | | | Water Catchment of the Boque River | | | Limit of the Regulations |
|---|---|---|---|---|---|---|---|---|---|---|
| Number of Sampling | S1 | S2 | S3 | S1 | S2 | S3 | S1 | S2 | S3 | Normative 0631/2015 [37] |
| pH | 7.36 | 7.49 | 7.33 | 7.55 | 6.05 | 6.57 | 7.61 | 7.56 | 7.63 | 6.0–9.0 |
| COD (mg/L) | 32.67 | <0.001 | 5.52 | 22.46 | <0.001 | 24.17 | 29.63 | 40.72 | 4.82 | 150 |
| Total solids (g/10 mL) | 0.0024 | 0.00165 | 0.0 | 0.0007 | 0.0058 | 0.0029 | 0.0001 | 0.00316 | 0.0005 | 50 |
| Cyanide (mg/L) | <0.025 | <0.025 | 0.025 | 1.02 | 1.32 | <0.025 | <0.025 | 1.57 | <0.025 | 1.0 |
| Cadmium (Cd) (mg/L) | 0.05 | 0.03 | 0.02 | $<1.0 \times 10^{-2}$ | 0.02 | 0.02 | $<1.0 \times 10^{-2}$ | 0.02 | 0.01 | 0.05 |
| Chrome (Cr) (mg/L) | $<1.0 \times 10^{-6}$ | $<1.0 \times 10^{-6}$ | $<1.0 \times 10^{-6}$ | 0.06 | $<1.0 \times 10^{-6}$ | 0.04 | $<1.0 \times 10^{-6}$ | $<1.0 \times 10^{-6}$ | $<1.0 \times 10^{-6}$ | 0.5 |
| Mercury (Hg) (mg/L) | 0.0008 | 0.0029 | 0.0003 | 0.001 | 0.0025 | 0.0008 | 0.0022 | 0.002 | 0.0008 | 0.002 |
| Nickel (Ni) (mg/L) | $<1.0 \times 10^{-3}$ | $<1.0 \times 10^{-3}$ | $<1.0 \times 10^{-3}$ | $<1.0 \times 10^{-3}$ | $<1.0 \times 10^{-3}$ | $<1.0 \times 10^{-3}$ | $<1.0 \times 10^{-3}$ | $<1.0 \times 10^{-3}$ | $<1.0 \times 10^{-3}$ | 0.5 |
| Zinc (Zn) (mg/L) | $<1.0 \times 10^{-3}$ | $<1.0 \times 10^{-3}$ | $<1.0 \times 10^{-3}$ | $<1.0 \times 10^{-3}$ | $<1.0 \times 10^{-3}$ | $<1.0 \times 10^{-3}$ | $<1.0 \times 10^{-3}$ | $<1.0 \times 10^{-3}$ | $<1.0 \times 10^{-3}$ | 3.0 |

mg/L: milligram per liter; S: sampling, the numbers 1S, 2S, and 3S correspond to the month of July, September, and December in which the sample was taken; <: less than the limit of quantification; n: is the number of samples; COD: chemical oxygen demand.

**Table A2.** Results of physicochemical parameters analysis compared with the normative 2115 of 2007.

| Physicochemical Parameters (n = 6) | House | | | Deep-Well Underground | | | Limit of the Regulations |
|---|---|---|---|---|---|---|---|
| Number of sampling | S1 | S2 | S3 | S1 | S2 | S3 | Normative 2115/2007 [38] |
| pH | 7.6 | 7.53 | 7.44 | 6.95 | 6.79 | 6.61 | 6.5–9.0 |
| Cyanide (mg/L) | <0.025 | 1.11 | <0.025 | <0.025 | <0.025 | <0.025 | 0.05 |
| Cadmium (Cd) (mg/L) | $<1.0 \times 10^{-2}$ | 0.03 | 0.03 | $<1.0 \times 10^{-2}$ | 0.01 | $<1.0 \times 10^{-2}$ | 0.003 |
| Chrome (Cr) (mg/L) | 0.02 | $<1.0 \times 10^{-6}$ | $<1 \times 10^{-6}$ | $<1.0 \times 10^{-6}$ | $<1.0 \times 10^{-6}$ | $<1.0 \times 10^{-6}$ | 0.05 |
| Mercury (Hg) (mg/L) | 0.0004 | 0.0005 | 0.0003 | 0.0007 | 0.0003 | 0.0007 | 0.001 |
| Nickel (Ni) (mg/L) | $<1.0 \times 10^{-3}$ | $<1.0 \times 10^{-3}$ | $<1.0 \times 10^{-3}$ | $<1.0 \times 10^{-3}$ | $<1.0 \times 10^{-3}$ | $<1.0 \times 10^{-3}$ | 0.2 |
| Zinc (Zn) (mg/L) | $<1.0 \times 10^{-3}$ | $<1.0 \times 10^{-3}$ | $<1.0 \times 10^{-3}$ | $<1.0 \times 10^{-3}$ | $<1.0 \times 10^{-3}$ | $<1.0 \times 10^{-3}$ | 3.0 |

mg/L: milligram per liter; S: sampling, the numbers 1S, 2S, and 3S correspond to the month of July, September, and December in which the sample was taken; <: less than the limit of quantification; n: is the number of samples.

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
