# Peer review of "Evaluation of the Effect of Gold Mining on the Water Quality in Monterrey, Bolívar (Colombia)"

_water, doi:10.3390/w12092523_

Round 1

Reviewer 1 Report

Paper is well written and all experiments are described well and clearly. What I see as problem is, that paper looks for me more as good technical report, then scientific paper. Described experiments are based on existing common technologies and also data analysis are standards. If paper will be consider like case study, than it could be accepted, but it is difficult to accept it as scientific paper.

Author Response

Dear Reviewer.

Thanks a lot for you evaluation.

Manuscript ID: water-908907

Title: Evaluation of Gold Mining Effect on the Water Quality in Monterrey, Bolivar (Colombia)

Authors: Alison Martín, Juliana Arias, Jennifer López, Lorena Santos, Camilo Venegas, Marcela Duarte, Nubia De Parra, Andrés Ortiz, Claudia Campos, Crispin Celis *

Reviewer 1

  1. Paper is well written and all experiments are described well and clearly. What I see as problem is, that paper looks for me more as good technical report, then scientific paper. Described experiments are based on existing common technologies and also data analysis are standards. If paper will be consider like case study, than it could be accepted, but it is difficult to accept it as scientific paper.

Reply: Dear Reviewer, thanks very much for your comment

This research generated data that contribute to the knowledge of the effects caused at the environmental and the public health by illegal and legal mining carried out with bad practices in countries with inefficient controls of this type of activities. The use of bioassays is recognized to evaluate toxicity effects, however, in this research; it is applied in an area sensitive to emerging countries where the use of these tests is limited or unknown. The results obtained suggest the use of animal and plant models to evaluate the effects on the environment and health of this type of activities and infer the damages that until now have not been sufficiently evaluated.  The authors believe that this article can be published in this special edition because it has coherence with the aim of the special issue, which is to emphasize different aspects and findings of the risk assessment for ecological and human health through soil, sediment and water contamination.

Similar paper published in this journal:

Gafur, N.; Sakakibara, M.; S.; Sano, S.; Sera, K. A Case Study of Heavy Metal Pollution in Water of Bone River by Artisanal Small-Scale Gold Mine Activities in Eastern Part of Gorontalo. Water 2018, 10, 1507.

Mercado-Garcia, R.; Beeckman, E. , Van Butsel, J.; Sanchez, M.; et al. Assessing the Freshwater Quality of a Large-Scale Mining Watershed: The Need for Integrated Approaches. Water 2019, 11, 1797.

Best regards.

Authors

Reviewer 2 Report

The title is clear.

The manuscript adheres to the journal's standards after revision

The key words permit found article in the current registers or indexes. Please put these in alphabetical order.

There are a lot of experimental data, relatively easy to read for person from other area.

The authors must underline the major findings of their work and explain how the use of their proposed procedures represents a progress to other similar published papers. Please point the novelty.

The Abstract must be revised. In abstract must be presents the study findings, methodologies, discussion as well as conclusion.

In the introduction isn’t clearly described the state of the art of the investigated problem.

The paper was written in standard, grammatically correct English, more corrections are necessary.

The materials, methods and equipment must be revised. The devices used for experiments must be presented.

The presentation not reflects the present state of knowledge. The research has been covered previously which was demonstrated by the references, but more citations from 2019-2020 are necessary. It is subject in actuality or no?

Please verify: The samples of water were diluted in four different concentrations 25, 50, 75 and 100% (v/v), reconstituted hard water was used for the animal models, while distilled water was used for the plant indicator. It is not clear!

The Conclusion must be revised. The conclusions weren’t been sufficiently justified. The Conclusion must contain major finding of experimental study.

The figure has good quality.

In Table 3. Mutagenic index, for each concentration analyzed in the five sampling stations with strain… please write numbers correct.

Please verify, Table 3 was introduced before Table 1 and Table 2.

The tables contain necessary results.

Please provide minimum 2 references from this journal (last years), for demonstrated that manuscript is in Waters topics.

Please respect guide for authors.

Author Response

Dear Reviewer Thanks a lot for your evaluation.

Manuscript ID: water-908907

Title: Evaluation of Gold Mining Effect on the Water Quality in Monterrey, Bolivar (Colombia)

Authors: Alison Martín, Juliana Arias, Jennifer López, Lorena Santos, Camilo Venegas, Marcela Duarte, Nubia De Parra, Andrés Ortiz, Claudia Campos, Crispin Celis *

Reviewer 2

  1. The title is clear.

Reply: Thanks.

  1. The manuscript adheres to the journal's standards after revision

Reply: Thank you very much dear Reviewer for your evaluation. It was prepared in accordance with the purpose of the special edition.

The authors' guide was revised, adjusting the article in relation to those established by the journal. The references were adjusted within the text, the punctuation of the numbers in Table 3, the bibliography was reviewed and adjusted, the grammar was increased and the keywords were placed in alphabetical order.

  1. The key words permit found article in the current registers or indexes. Please put these in alphabetical order.

Reply:

The keywords were organized alphabetically (Line N 31).

  1. There are a lot of experimental data, relatively easy to read for person from other area.

Reply: Thanks very much

  1. The authors must underline the major findings of their work and explain how the use of their proposed procedures represents a progress to other similar published papers. Please point the novelty.

Reply:

The most relevant results of this study are related to the use of biological models that can be used as indicators of toxicity in waters contaminated by gold mining residues, including mercury. Bioassays are a very useful tool to demonstrate the effects on the environment and on human health. These assays help to the sanitation organizations in the different countries to take preventive actions on this issue. Given the difficulties of being able to measure these effects, not only due to the remoteness of the areas where mining occurs but also due to economic limitations. Having easy to perform, cheap, and reliable analyzes, these become a useful tool to monitor and apply corrective measures in similar situations to the one proposed in this study. On the other hand, the study corroborates the importance of jointly carrying out physical-chemical analyzes and bioassays to make a better evaluation of the effects of gold mining. The importance of including bioassays to assess the health risk and as a warning system when concentrations of contaminants cannot be detected but are present and affect human health in the long term is highlighted. All these aspects were highlighted in the conclusions section (Line N 431-451)

  1. The Abstract must be revised. In abstract must be presents the study findings, methodologies, discussion as well as conclusion.

Reply:

The abstract was reviewed, and the corresponding adjustments were made. In this, the objective of the study, main results and conclusions were emphasized. (Lines N 17-30). In the other hand, we have oly 200 word for describe all the paper and we must be concreted.

  1. .In the introduction isn’t clearly described the state of the art of the investigated problem.

Reply:

The introduction was revised and adjusted, as well as new references were included. With the additions the state of the art is clarified, highlighting its importance and novelty (Line N 38-40, 43-46, 49-51, 52-55, 72-74, 85-89 and  92-108).

  1. The paper was written in standard, grammatically correct English, more corrections are necessary.

Reply:

The English was revised; some changes were made in the grammar or rewriting to make it clearer and the English to improve.

  1. The materials, methods and equipment must be revised. The devices used for experiments must be presented.

Reply:

The materials and methods were reviewed, and the equipment (s) used for this type of analysis were added for the physicochemical analysis. For the microbiological tests, the name and brand of the main culture media / main reagents were added, likewise for some methods the brand of some equipment was specified. The changes are on the following lines: 120-126, 138-143, 157-158, 161-162, 173-174, 176-177 and 183-185.

The presentation not reflects the present state of knowledge. The research has been covered previously which was demonstrated by the references, but more citations from 2019-2020 are necessary. It is subject in actuality or no?

Reply:

The document was revised and adjusted, as well as more recent references included from 2019 to 2020. The added references allow to give more support to the results allowing to argue the results obtained in the study. Changes in the introduction can be seen in yellow color and new references included throughout the document are seen with the following numbering: 4, 6, 28, 30, 55, 62, 63, 73, 78 and 82 (bibliography section).

  1. Please verify: The samples of water were diluted in four different concentrations 25, 50, 75 and 100% (v/v), reconstituted hard water was used for the animal models, while distilled water was used for the plant indicator. It is not clear!

Reply:

The sentence was rewritten to make it clearer for the reader and to understand the procedure performed during the bioassay analysis. The changes are seen on Lines 137-140.

  1. The Conclusion must be revised. The conclusions weren’t been sufficiently justified. The Conclusion must contain major finding of experimental study.

Reply:

The conclusions were revised, and rewritten. New aspects were added to make the conclusions clearer. Changes are seen on lines 431-451.

  1. The figure has good quality.

Reply: Thanks very much.

  1. In Table 3. Mutagenic index, for each concentration analyzed in the five sampling stations with strain… please write numbers correct.

Reply:

The numbers in Table 3 were revised and adjusted according to the authors' guide. The changes are directly reflected in Table 3.

  1. Please verify, Table 3 was introduced before Table 1 and Table 2.

Reply:

The mention of table 3 was deleted in line 114. This was due to an error during the article editing process. Table 3 is correctly mentioned in the lines 234 and 237.

  1. The tables contain necessary results.

Reply: Thanks very much

  1. Please provide minimum 2 references from this journal (last years), for demonstrated that manuscript is in Waters topics.

Reply:

The search for articles from Water journal and another important journal were preceded directly on the journal's website. Articles related to the research topics were found and they were included throughout the article. These were introduced both in the introduction and in the discussion. The new references included are throughout the document and are seen with the following numbering: 3, 55, 61 and 76 (bibliography section).

Gafur, N.; Sakakibara, M.; S.; Sano, S.; Sera, K. A Case Study of Heavy Metal Pollution in Water of Bone River by Artisanal Small-Scale Gold Mine Activities in Eastern Part of Gorontalo. Water 2018, 10, 1507.

Mercado-Garcia, R.; Beeckman, E. , Van Butsel, J.; Sanchez, M.; et al. Assessing the Freshwater Quality of a Large-Scale Mining Watershed: The Need for Integrated Approaches. Water 2019, 11, 1797.

  1. Please respect guide for authors.

Reply:

The guides for authors were revised, adjusting the article in relation to those established by the journal. Allowing the following changes to be made: the references were adjusted within the text, the punctuation in the figures or numbers described in the results tables was reviewed and the bibliography was adjusted and the keywords were placed in alphabetical order

  1. I recommend MAJOR REVISION, there are a lot of interesting experimental data, but manuscript must be revised and improved.

Reply:

The article was revised proceeding with the suggested corrections and adjustments. Likewise, aspects of the article were improved, such as the summary, introduction, discussion, and conclusions.

Reviewer 3

  1. The authors should clearly explain in the Introduction what is the legal status and how the environmental permits are allocated for the gold mining process in Colombia. Is the illegal gold mining reported as " growth of the gold metal production with 7% (page 1)? How the illegal practices may be maintained when chemicals such as cyanide and mercury are used in the gold extraction process? These are not usual chemicals that may be purchased without problems.

Reply:

It was clarified what is related to the phrase "growth of gold metal production with 7%. Which corresponds to a general amount in relation to total production produced within Colombia and the origin of mining is not differentiated. On the other hand, n countries with strict regulations and controls on gold mining, the use of substances such as cyanide and mercury could be avoided. However, in Colombia and other emerging countries, illegal exploitation is relevant and there are no effective controls. This kind of study allows government entities to take corrective measures. Colombia is one of the most important gold producers, as well as a great consumer of mercury for mining activities. Having previously evaluated animal and plant models offers an inexpensive and reliable tool that can be useful to inform and control these activities. The data obtained and the protocols used in this study can be applied in regions with similar situations. Parallel to the illegal mining activity is the illicit commercialization of equipment and supplies for mining.

  1. Normally the environmental permits contain also discharge limits or norms /standards for the wastewater discharged from industrial processes. This should be also the case of gold mining activities and the authors should clarify what is the legal status about these wastewater discharges for the studied area.

Reply:

In this case, two factors are important. 1. Although there are regulations for the control of water discharges from gold mining, the limits that are established are related to the detection limit of the instrument according to its technology and based on the maximum concentration that can be removed by an effective treatment system. This is not related to the traces that we are not able to measure but that over time accumulate in tissues of animals, plants, and animals, causing damage to the environment and human health. Hence the importance of using bioassays as alert systems as they are more sensitive to the levels of contamination present in the water. 2. The practice is illegal because it does not have the permits and previous studies of environmental licenses that the government must issue. Neither are there adequate practices for the use of chemicals, or treatment systems. These practices are common in many parts of the world where mining is done.

  1. The study objectives should be clearly stated as well as the novelty of this article (comparing it to other articles published in the same field by this journal or other international journals). Physical chemical monitoring and bioassays are not new research topics and this is why the clear statement of the novelty is important.

Reply:

The most relevant results in this study are related to the evaluation of the models that can be used as indicators of toxicity in waters contaminated by residues from gold mining. These bioassays become very useful tools to demonstrate effects at the environmental level and on human health in situations like those in Colombia. These are rivers that receive exclusive contamination from mining, far from urban areas and other activities. Faced with the difficulties of being able to measure these effects, not only because of the remoteness of the areas, but also due to economic limitations, having easy-to-perform, cheap and reliable analyzes offers useful tools for monitoring and applying corrective measures in situations similar to the one proposed in this study.

On the other hand, it corroborates the importance of jointly carrying out physical, chemical and bioassay analyzes to make a better evaluation of the effects of gold mining. The importance of including bioassays to assess sanitary risk and as an alert system when concentrations of pollutants cannot be detected but are present and affect human health in the long term, is highlighted.

The authors believe that this article can be published in this special issue because it has coherence with the aim of the edition, which is to emphasize different aspects and findings of the risk assessment for ecological and human health through soil, sediment and water contamination.

  1. The authors should describe in the Materials and methods section the equipment and procedures for the analysis of all water quality indicators (physical, chemical, microbiological, bioassays).

Reply:

In the method materials part, the equipment used for the detection and reading of results was included, whether for the chemical, microbiological or bioassay part. Likewise, the brands of the culture media and reagents used for the detection of microorganisms and bioassays and essential conditions for their performance are listed, such as the average concentration of water hardness for bioassays. Lines 120-126, 138-143, 157-158, 161-162, 173-174 and 176-177.

  1. The major problem of this study is the fact that the water samples collection in various parts of the rivers and the wastewater discharges are not correlated so as to give an adequate dimension of the pollution problem. For such an approach a different way of sampling would have been needed and many other issues should have been considered such as the river water flow, precipitations, other polluting discharges, distance from the discharge points, etc.

Reply:

We agree with these suggestions. However, the economic resources were limited, the impact zone quite remote and difficult to access, with some civil security risks. Under these conditions, the analysis of the water was privileged and impact results were obtained. This allowed us to highlight the risk to the population health and the need to control the discharge of pollutants into the water. An important point to highlight is that these results were presented to government agencies and recently a water purification system was approved for the population of Monterrey. The authors want to point out that a more exhaustive sampling system in the river to evaluate the variables cited by the evaluator should be carried out with special care for the researchers because it is an area dominated by the guerrillas or groups outside the law.

  1. Most of the references are quite old. The authors should improve their status by selecting relevant new studies.

Reply:

A new search of the bibliography was carried out to update and complement them. Articles related to the research topics were found and they were included throughout the article. These were introduced both in the introduction and in the discussion. The new references included throughout the document are seen with the following numbering: 3, 4, 6, 7, 28, 30, 63, 73, 75, 76, 78 and 82 (bibliography section).

Reviewer 3 Report

The manuscript "Evaluation of gold mining effect on the water quality in Monterrey, Bolivar (Colombia)" is generally well written and structured. Before being accepted for publication the authors should consider the following issues to improve it, my recommendation being Major revision.

  1. The authors should clearly explain in the Introduction what is the legal status and how the environmental permits are allocated for the gold mining process in Colombia. Is the illegal gold mining reported as " growth of the gold metal production with 7% (page 1) ? How the illegal practices may be maintained when chemicals such as cyanide and mercury are used in the gold extraction process? These are not usual chemicals that may be purchased without problems.
  2. Normally the environmental permits contain also discharge limits or norms /standards for the wastewater discharged from industrial processes. This should be also the case of gold mining activities and the authors should clarify what is the legal status about these wastewater discharges for the studied area.
  3. The study objectives should be clearly stated as well as the novelty of this article (comparing it to other articles published in the same field by this journal or other international journals). Physical chemical monitoring and  bioassays are not new research topics and this is why the clear statement of the novelty is important.
  4. The authors should describe in the Materials and methods section the equipment and procedures for the analysis of all water quality indicators (physical, chemical, microbiological, bioassays)
  5. The major problem of this study is the fact that the water samples collection in various parts of the rivers and the wastewater discharges are not correlated so as to give an adequate dimension of the pollution problem. For such an approach a different way of sampling would have been needed and many other issues should have been considered such as the river water flow, precipitations, other polluting discharges, distance from the discharge points, etc
  6. Most of the references are quite old. The authors should improve their status by selecting relevant new studies.

Author Response

Dear Reviewer thanks a lot for your evaluation.

Manuscript ID: water-908907

Title: Evaluation of Gold Mining Effect on the Water Quality in Monterrey, Bolivar (Colombia)

Authors: Alison Martín, Juliana Arias, Jennifer López, Lorena Santos, Camilo Venegas, Marcela Duarte, Nubia De Parra, Andrés Ortiz, Claudia Campos, Crispin Celis *

Reviewer 3

  1. The authors should clearly explain in the Introduction what is the legal status and how the environmental permits are allocated for the gold mining process in Colombia. Is the illegal gold mining reported as " growth of the gold metal production with 7% (page 1)? How the illegal practices may be maintained when chemicals such as cyanide and mercury are used in the gold extraction process? These are not usual chemicals that may be purchased without problems.

Reply:

It was clarified what is related to the phrase "growth of gold metal production with 7%. Which corresponds to a general amount in relation to total production produced within Colombia and the origin of mining is not differentiated. On the other hand, n countries with strict regulations and controls on gold mining, the use of substances such as cyanide and mercury could be avoided. However, in Colombia and other emerging countries, illegal exploitation is relevant and there are no effective controls. This kind of study allows government entities to take corrective measures. Colombia is one of the most important gold producers, as well as a great consumer of mercury for mining activities. Having previously evaluated animal and plant models offers an inexpensive and reliable tool that can be useful to inform and control these activities. The data obtained and the protocols used in this study can be applied in regions with similar situations. Parallel to the illegal mining activity is the illicit commercialization of equipment and supplies for mining.

  1. Normally the environmental permits contain also discharge limits or norms /standards for the wastewater discharged from industrial processes. This should be also the case of gold mining activities and the authors should clarify what is the legal status about these wastewater discharges for the studied area.

Reply:

In this case, two factors are important. 1. Although there are regulations for the control of water discharges from gold mining, the limits that are established are related to the detection limit of the instrument according to its technology and based on the maximum concentration that can be removed by an effective treatment system. This is not related to the traces that we are not able to measure but that over time accumulate in tissues of animals, plants, and animals, causing damage to the environment and human health. Hence the importance of using bioassays as alert systems as they are more sensitive to the levels of contamination present in the water. 2. The practice is illegal because it does not have the permits and previous studies of environmental licenses that the government must issue. Neither are there adequate practices for the use of chemicals, or treatment systems. These practices are common in many parts of the world where mining is done.

  1. The study objectives should be clearly stated as well as the novelty of this article (comparing it to other articles published in the same field by this journal or other international journals). Physical chemical monitoring and bioassays are not new research topics and this is why the clear statement of the novelty is important.

Reply:

The most relevant results in this study are related to the evaluation of the models that can be used as indicators of toxicity in waters contaminated by residues from gold mining. These bioassays become very useful tools to demonstrate effects at the environmental level and on human health in situations like those in Colombia. These are rivers that receive exclusive contamination from mining, far from urban areas and other activities. Faced with the difficulties of being able to measure these effects, not only because of the remoteness of the areas, but also due to economic limitations, having easy-to-perform, cheap and reliable analyzes offers useful tools for monitoring and applying corrective measures in situations similar to the one proposed in this study.

On the other hand, it corroborates the importance of jointly carrying out physical, chemical and bioassay analyzes to make a better evaluation of the effects of gold mining. The importance of including bioassays to assess sanitary risk and as an alert system when concentrations of pollutants cannot be detected but are present and affect human health in the long term, is highlighted.

The authors believe that this article can be published in this special issue because it has coherence with the aim of the edition, which is to emphasize different aspects and findings of the risk assessment for ecological and human health through soil, sediment and water contamination.

  1. The authors should describe in the Materials and methods section the equipment and procedures for the analysis of all water quality indicators (physical, chemical, microbiological, bioassays).

Reply:

In the method materials part, the equipment used for the detection and reading of results was included, whether for the chemical, microbiological or bioassay part. Likewise, the brands of the culture media and reagents used for the detection of microorganisms and bioassays and essential conditions for their performance are listed, such as the average concentration of water hardness for bioassays. Lines 120-126, 138-143, 157-158, 161-162, 173-174 and 176-177.

  1. The major problem of this study is the fact that the water samples collection in various parts of the rivers and the wastewater discharges are not correlated so as to give an adequate dimension of the pollution problem. For such an approach a different way of sampling would have been needed and many other issues should have been considered such as the river water flow, precipitations, other polluting discharges, distance from the discharge points, etc.

Reply:

We agree with these suggestions. However, the economic resources were limited, the impact zone quite remote and difficult to access, with some civil security risks. Under these conditions, the analysis of the water was privileged and impact results were obtained. This allowed us to highlight the risk to the population health and the need to control the discharge of pollutants into the water. An important point to highlight is that these results were presented to government agencies and recently a water purification system was approved for the population of Monterrey. The authors want to point out that a more exhaustive sampling system in the river to evaluate the variables cited by the evaluator should be carried out with special care for the researchers because it is an area dominated by the guerrillas or groups outside the law.

  1. Most of the references are quite old. The authors should improve their status by selecting relevant new studies.

Reply:

A new search of the bibliography was carried out to update and complement them. Articles related to the research topics were found and they were included throughout the article. These were introduced both in the introduction and in the discussion. The new references included throughout the document are seen with the following numbering: 3, 4, 6, 7, 28, 30, 63, 73, 75, 76, 78 and 82 (bibliography section).

Round 2

Reviewer 1 Report

Dear Authors, thanks for explanation and update. Looks good for me

Author Response

Dear Reviewer Thanks very much for your time.

Reviewer 2 Report

The title is clear.

The authors affiliation order was corrected.

The manuscript adheres to the journal's standard.

The key words permit found article in the current registers or indexes, the authors ordered these alphabetically.

There are a lot of experimental data, relatively easy to read for person from other area.

The authors underlined the major findings of their work and explain how the use of their proposed procedures represents a progress.

The Abstract was rewrite. In abstract was presented the study findings, methodologies, discussion as well as conclusion.

In the introduction is clearly described the state of the art of the investigated problem.

The paper was written in standard, grammatically correct English. More corrections were made.

The materials, methods and equipment were revised. The devices used for experiments were presented.

The presentation reflects the present state of knowledge.

The Conclusion was revised. The figure has good quality.

Tables contain necessary experimental data, these were improved.

Please respect guide for authors when write the references, these aspect can be corrected in proofing process. There are references with journal abbreviation or without abbreviation.

Author Response

Dear Reviewer, Thanks very much for your comments.

  • Please respect guide for authors when write the references, these aspect can be corrected in proofing process. There are references with journal abbreviation or without abbreviation.

Reply:

Authors review each of the references and adjust the journal titles in their abbreviated form. Changes are in red at the bibliography section according to guide for authors.

Reviewer 3 Report

As the result of the review process the MS has been improved. However, few minor suggestions for corrections are addressed below:

Line 88_"of potentially carcinogenic or mutant chemicals "- replace mutant with mutagenic

Line 103 and not only " ...by bad mining exploitation practices ..." replace by bad exploitation mining practices

Author Response

Dear Rewiewer, thanks for your time .

  • Line 88_"of potentially carcinogenic or mutant chemicals "- replace mutant with mutagenic

Reply:

Thanks for your suggestion. The word in line 88 was replaced as follows: mutant with mutagenic. The change is in red at line 88.

  • Line 103 and not only " ...by bad mining exploitation practices ..." replace by bad exploitation mining practices.

Reply:

Thanks for your suggestion. The words on line 103 are now: ¨bad exploitation mining practices¨

Best regards.

Authors.